# NeurIPS 2023 Competition: Privacy Preserving Federated Learning Document VQA

**Marlon Tobaben**[1][*]   **Mohamed Ali Souibgui**[2]   **Rubèn Tito**[2]   **Khanh Nguyen**[2]
**Raouf Kerkouche**[3]   **Kangsoo Jung**[4]   **Joonas Jälkö**[1]   **Lei Kang**[2]   **Andrey Barsky**[2]
**Vincent Poulain d'Andecy**[5]   **Aurélie JOSEPH**[5]   **Aashiq Muhamed**[6]   **Kevin Kuo**[6]
**Virginia Smith**[6]   **Yusuke Yamasaki**[7]   **Takumi Fukami**[7]   **Kenta Niwa**[7]   **Iifan Tyou**[7]
**Hiro Ishii**[8]   **Rio Yokota**[8]   **Ragul N**[9]   **Rintu Kutum**[9]   **Josep Llados**[2]
**Ernest Valveny**[2]   **Antti Honkela**[1]   **Mario Fritz**[3]   **Dimosthenis Karatzas**[2]

[1]University of Helsinki   [2]Computer Vision Center, Universitat Autònoma de Barcelona
[3]CISPA Helmholtz Center for Information Security   [4]INRIA   [5]Yooz
[6]Carnegie Mellon University   [7]NTT   [8]Tokyo Institute of Technology   [9]Asoka University

## Abstract

The Privacy Preserving Federated Learning Document VQA (PFL-DocVQA) competition challenged the community to develop provably private and communication-efficient solutions in a federated setting for a real-life use case: invoice processing. The competition introduced a dataset of real invoice documents, along with associated questions and answers requiring information extraction and reasoning over the document images. Thereby, it brings together researchers and expertise from the document analysis, privacy, and federated learning communities. Participants fine-tuned a pre-trained, state-of-the-art Document Visual Question Answering model provided by the organizers for this new domain, mimicking a typical federated invoice processing setup. The base model is a multi-modal generative language model, and sensitive information could be exposed through either the visual or textual input modality. Participants proposed elegant solutions to reduce communication costs while maintaining a minimum utility threshold in track 1 and to protect all information from each document provider using differential privacy in track 2. The competition served as a new testbed for developing and testing private federated learning methods, simultaneously raising awareness about privacy within the document image analysis and recognition community. Ultimately, the competition analysis provides best practices and recommendations for successfully running privacy-focused federated learning challenges in the future.

## 1   Introduction

Automatic document image processing has become a highly active research field in recent years [Appalaraju et al., 2024, Lee et al., 2023, Tito et al., 2023a], with invoices being one of the most frequently processed document types [Šimsa et al., 2023]. In a typical real-life invoicing scenario, business suppliers produce invoices for their services and send them to their customers. These documents contain sensitive information, such as consumer/purchaser identity, transaction details, purpose, date, phone numbers, amount paid, account information for payment, etc. The customers (document users) need

---

[*]This analysis is jointly written by organizers and participants. See author contributions in Appendix A.1.

Submitted to the 38th Conference on Neural Information Processing Systems (NeurIPS 2024) Track on Datasets and Benchmarks. Do not distribute.

to extract this information and take the corresponding actions (i.e. reject, or make a payment against the invoice). In automated pipelines, these documents would be sent to AI technology providers, typically offered in the form of cloud services[2], which automatically extract all required information from the documents, and return it to the document users.

A generic approach to extract information from invoices is DocVQA [Mathew et al., 2020]. The extraction is done by asking questions in a natural language form to get specific information as answers, using a deep learning model. However, training an accurate DocVQA model requires a considerable amount of data, that is rarely held by a single entity. One solution is to train this model collaboratively by aggregating and centralizing data from a set of clients that face the same problem. But, documents often cannot be freely exchanged due to the sensitive information they contain. Federated Learning (FL) is a learning paradigm that purports to solve this problem [McMahan et al., 2017b]. Rather than exchanging privately-held data, participating entities (known as clients) train models on their data in a decentralized fashion, exchanging only the local model updates with a central server. However, even though FL is more private than the centralized approach, a significant amount of information can still be inferred from the updates shared during training, or from the parameters of the resulting trained model, whether by an adversarial server, client, or downstream user [Sikandar et al., 2023].

Differential Privacy (DP) [Dwork et al., 2016] is considered the gold standard in terms of privacy preservation and can be used to provide provable privacy guarantees. DP formally quantifies the maximum information leakage from the inclusion of any one individual record in a dataset. Deep learning models can be trained under DP by clipping parameter updates and adding noise to them [Rajkumar and Agarwal, 2012, Song et al., 2013, Abadi et al., 2016]. However, this introduces a trade-off between privacy and utility. Stronger privacy guarantees require introducing more noise, which proportionately degrades model accuracy.

Another drawback of FL is the high communication cost [Kairouz et al., 2021]. At each federated round, the global model is transmitted by the server to selected clients (downstream step) to be trained on their local data, and then the update of this model is sent by these selected entities back to the server (upstream step). For models with millions or even billions of parameters, this requires significant bandwidth, multiplied by the number of federated rounds required to reach model convergence.

In this paper, we present an analysis of the NeurIPS 2023 competition on privacy preserving FL DocVQA that we designed to expose the above challenges and invite the community to design novel creative solutions for this real-life use case. It brought together researchers and expertise from the document analysis, privacy, and FL communities. Additionally, it added a realistic use case for privacy and FL researchers as well as expanding the scope of document analysis to DP solutions.

## 2   Related Work

**Document Visual Question Answering (DocVQA)** DocVQA has been an evolving field during the last few years. This is due to the emerging datasets that address different document domains. For instance, industry documents [Mathew et al., 2020, 2021, Tito et al., 2021b, 2023a], infographics [Mathew et al., 2022], multidomain [Landeghem et al., 2023a,b], open-ended questions [Tanaka et al., 2021], multilingual [Qi et al., 2022], multipage [Tito et al., 2023a] or collections of documents [Tito et al., 2021a]. However, these datasets are often small and highly domain-specific, which limits generalizability.

**Federated Learning (FL)** FL [Shokri and Shmatikov, 2015, McMahan et al., 2017b] addresses this issue, and has seen practical use in both research and industrial applications [Li et al., 2020], particularly in domains where sensitive data is common, such as medicine [Dayan et al., 2021] and finance [Long et al., 2020]. FL carries a trade-off between model utility, data privacy, and communication efficiency [Zhang et al., 2023], and requires specific consideration of client data heterogeneity,

---

[2]Automatic document processing services offered by large corporations (AWS Intelligent Document Processing, Google Cloud Document AI, Microsoft Azure Form Recognizer, etc) or specialized providers.

scalability, and fault tolerance. Much recent work in FL focuses on mitigating these problems, primarily through developments in aggregation algorithms [Moshawrab et al., 2023, Elkordy and Avestimehr, 2022, So et al., 2022, Nguyen et al., 2022], but also in parameter compression [Tang et al., 2019] and quantization [Xu et al., 2022].

**Privacy Attacks** While FL offers privacy advantages, it remains vulnerable to various attacks that jeopardize client dataset privacy. In the federated architecture, both the server and clients can potentially act as adversaries. Gradient updates in FL have the potential to disclose information about the training data, making them susceptible to "gradient inversion attacks" [Zhu et al., 2019, Zhao et al., 2020, Fu et al., 2022, Wainakh et al., 2022, Li et al., 2022b, Geiping et al., 2020, Melis et al., 2019, Li et al., 2022d], which enable accurate data reconstruction. Moreover, adversaries can execute "membership inference attacks" [Nasr et al., 2019, Melis et al., 2019, Suri et al., 2022, Shokri et al., 2017, Choquette-Choo et al., 2021, Li and Zhang, 2021, Hu et al., 2022b] to infer the inclusion of specific data points in other participants' datasets, as well as "property inference attacks" [Melis et al., 2019] to deduce subgroup statistics despite secure aggregation [Kerkouche et al., 2023, Pejó and Biczók, 2023]. FL inherently lacks protection against these threats, necessitating explicit mitigation strategies to safeguard client data from adversaries.

**Differential Privacy (DP)** $(\epsilon, \delta)$-DP [Dwork et al., 2006] has a privacy budget consisting of $\epsilon \geq 0$ and $\delta \in [0, 1]$, where smaller values correspond to a stronger privacy guarantee. Especially relevant to our setting is group-level DP, which preserves privacy leakage from the inclusion or exclusion of groups of datapoints [Galli et al., 2023, Marathe and Kanani, 2022], such as multiple records associated with a specific user. We refer to Dwork and Roth [2014] for a comprehensive intro to DP.

**High utility models under DP** Currently, many works improve the utility-privacy trade-off through transfer learning [Yosinski et al., 2014] assuming the availability of non-sensitive public data for pre-training and only utilizing DP to protect sensitive downstream data during fine-tuning. We would like to refer to Tramèr et al. [2022a] for a discussion on the drawbacks of these assumptions. Transfer learning is highly effective for both language [Li et al., 2022c, Yu et al., 2022a] and vision tasks [Cattan et al., 2022, De et al., 2022, Kurakin et al., 2022, Tobaben et al., 2023]. In particular, parameter-efficient fine-tuning [Houlsby et al., 2019] with adaptation methods such as LoRA [Hu et al., 2022a] have been demonstrated to yield improved utility-privacy trade-offs for DP, as have quantization [Youn et al., 2023] or compression of model updates [Kerkouche et al., 2021a,b, Miao et al., 2022]. All these methods reduce the size of the updates, and thereby reduce the amount of noise addition required. The same strategies often yield competitive performance for FL.

# 3    General Competition Information

This section describes general information about the competition that is common to both tracks. These are the dataset, metrics, model, starter kit and the participation statistics.

## 3.1    PFL-DocVQA Dataset

For this competition, we created PFL-DocVQA [Tito et al., 2023b], the first dataset for private federated DocVQA. The dataset is created using invoice document images gathered from the DocILE dataset [Šimsa et al., 2023]. For every image, we provide the OCR transcription and form a set of question/answer pairs. The competition's version of PFL-DocVQA contains a total of 336,842 question-answer pairs framed on 117,661 pages of 37,669 documents from 6,574 different invoice providers. PFL-DocVQA is designed to be used in two tasks, and so is divided into two subsets. For the first task of training and evaluating machine learning privacy-preserving solutions on DocVQA in a FL fashion, a base subset of PFL-DocVQA called the "BLUE" data is used. In the second task, membership inference attacks are designed to assess the privacy guarantees of the DocVQA models that were trained with the base data. These attacking approaches are to utilize a second subset called the "RED" data. In this competition, we focus on the first task, thus, we use only the "BLUE" data. For more details on the full PFL-DocVQA datasets, refer to Tito et al. [2023b].

PFL-DocVQA aims to train and evaluate DocVQA systems that protect sensitive document information. In our scenario, sensitive information encompasses all information originating from each invoice provider. Therefore, an effective model must prevent the disclosure of any details associated with these providers (such as provider names, emails, addresses, logos, etc.) across diverse federated clients. Following this, the base data used in this competition consists of a training set divided among $N$ clients (we use $N = 10$), a validation set and a test set. (See Figure A.1). The training set of each of the $N$ clients contains invoices sampled from a different subset of providers, resulting in a highly non-i.i.d. distribution. In the validation and test sets, we include documents both from the providers that were seen during training, and from a set of providers that were not seen, to better evaluate the generalizability of the models.

## 3.2 Evaluation Metrics

In the PFL-DocVQA Competition three main aspects are evaluated: The model's utility, the communication cost during training and the DP privacy budget spent through training the model.

**Utility** To evaluate the visual question answering performance of the participants' methods we use accuracy and ANLS (Average Normalized Levenshtein Similarity), a standard soft version of accuracy extensively used in most of the text-based VQA tasks [Biten et al., 2019a,b, Mathew et al., 2020, Tito et al., 2021b, Mathew et al., 2021, Tito et al., 2021a, Mathew et al., 2022, Tito et al., 2023a, Landeghem et al., 2023b,a]. This metric is based on the normalized Levenshtein Distance [Levenshtein, 1966] between the predicted answer and the ground truth, allowing us to assess the method's reasoning capabilities while smoothly penalizing OCR errors.

**Communication cost** We measure the efficiency of the communications as the total amount of information transmitted between the server and the clients in Gigabytes (GB) in both directions. The initial transmission of the pre-trained model to the clients is not included in the communication cost.

**Privacy** The methods of track 2 are required to comply with a DP privacy budget of no more than a pre-defined $\epsilon \in \{1, 4, 8\}$ at $\delta = 10^{-5}$. We provided a script within the starter kit detailed in Section 3.4 to compute the required noise multiplier given the target ($\epsilon, \delta$). Participants may need to adjust the script to their algorithms. Moreover, we required the participants to upload a theoretical privacy proof of their methods, which was manually reviewed by the competition organizers.

## 3.3 Pre-trained Model

The participants were asked to implement their solutions starting from the same pre-trained model. The architecture chosen is Visual T5 (VT5), it is a multimodal generative network consisting of a simplified version of Hi-VT5 [Tito et al., 2023a], which was originally proposed for multi-page DocVQA. VT5 exploits the image and text modalities, which is beneficial to perform the DocVQA task. However, this dual-modality approach also presents a more complex challenge: safeguarding private information across both modalities, compared to handling just one. Moreover, VT5 is a generative model based on the T5 [Raffel et al., 2020] language model. Language models can suffer hallucinations [Rawte et al., 2023], leading to the potential leakage of private information.

The architecture VT5 consists of an encoder-decoder model based on T5. The input of the model is the question, the OCR tokens of the document (text and spatial information), and the encoded document image using the DiT [Li et al., 2022a] vision transformer model. These three inputs are concatenated and fed to the VT5 to output the answer following the autoregressive mechanism.

We also provide pre-trained weights for VT5. First, the language backbone T5 is initialized with the pre-trained weights on the C4 dataset [Raffel et al., 2020], and the visual DiT with the pre-trained weights on the document classification task. After that, the full model is fine-tuned on the single-page DocVQA task, using the SP-DocVQA dataset [Mathew et al., 2020, 2021] for 10 epochs.

Table 1: Participation Statistics as of May 31, 2024.

| Registrations to platform | Downloads | Countries | Submissions Track 1 | Submissions Track 2 |
|---|---|---|---|---|
| 382 | 494 | 21 | 13 | 6 |

## 3.4 Starter Kit

The starter kit includes the pre-trained model checkpoint, the fine-tuning dataset, code for running the baselines and instructions on how to run and modify the code. The code itself is based on established libraries such as PyTorch [Paszke et al., 2019] and the FL framework Flower [Beutel et al., 2020]. Besides the training code, the starter kit includes functions for computing the privacy parameters based on the hyperparameters and for logging the communication between server and clients. We tested the installation and execution of the baseline on various clusters across different institutions and provided support to participants if they encountered any difficulties. The starter kit is openly available: `https://github.com/rubenpt91/PFL-DocVQA-Competition`.

## 3.5 Participation Statistics

Refer to Table 1 for the participation statistics. Our competition has gained interest across the communities and remains an open benchmark in the future: `https://benchmarks.elsa-ai.eu/?ch=2&com=introduction`. In Section 6.2 we discuss measures to lower the participation threshold.

# 4 Track 1: Communication Efficient Federated Learning

Track 1 focuses on training high utility models while reducing the communication cost in federated learning. We describe the task, the organizers' baseline and two submitted approaches (See Table 2).

## 4.1 Task Formulation

The objective of track 1 is to reduce the communication used (# bytes), while achieving a comparable utility (ANLS) with the organizers' baseline. The baseline achieved a validation ANLS of 0.8676 and we define a comparable utility to the baseline as 0.8242 ANLS (5% w.r.t. the baseline). Any submission that achieves at least that ANLS is valid, thus the deciding factor for winning the competition is the communication efficiency, which is measured using a single metric. We opted for scoring using a single metric as the trade-off between utility and communication is not linear. Furthermore, in real world applications less communication efficiency will lead to higher monetary costs or longer training times that need to be considered in contrast to changes in model utility.

Participants are required to use the VT5 baseline model with the initial pre-trained weights and utilize only the PFL-DocVQA dataset for fine-tuning. Further the participants are not allowed to change the PFL-DocVQA data distribution. Additionally, participants are required to upload a log of the communication between the clients and the central party (# bytes) and the final model checkpoint.

The organizers evaluate the model utility on a secret test set and thus the model architecture needs to be the same as the initial baseline. While this makes some solutions such model distillation more challenging, the track is open to a wide range of possible solutions. Participants could, e.g., utilize parameter-efficient fine-tuning, compression of the FL updates, lower precision or better hyper-parameters to achieve higher communication efficiency while maintaining a comparable utility.

## 4.2 Baseline Solution Track 1

The baseline solution for track 1 fine-tunes all parameters of the pre-trained model but the visual module. It essentially uses Federated Averaging (FedAvg) [McMahan et al., 2017a]. In each global round, the central server samples $K = 2$ clients out of all $N = 10$, and each of these clients computes

Table 2: Competition Winners Track 1 (Communication efficient federated learning)

| Rank | Team | Method | Communication ↓ | ANLS ↑ |
|------|------|--------|-----------------|--------|
| 1 | Muhamed et al. (Section 4.3) | LoRA | 0.38 GB (-99.14%) | 0.8566 (-3.45%) |
| 2 | Niwa et al. (Section 4.4) | FedShampoo | 10.01 GB (-77.37%) | 0.8891 (+0.20%) |
| - | Organizers (Section 4.2) | Baseline | 44.65 GB | 0.8873 |

the weight updates locally across multiple local rounds. The central server aggregates the client updates and communicates the updated model to the sampled clients in the next round.

## 4.3 Winner Track 1: Muhamed, Kuo, and Smith

We considered three orthogonal methods to reduce communication (LoRA, tuning FL hyperparameters, and quantization). The winning solution for Track 1 uses only LoRA ($100\times$ reduction). Combining all methods can achieve a $5200\times$ reduction. For complete details, see Appendix C.

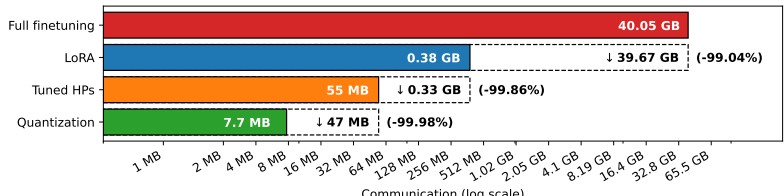

**1. LoRA. Lo**w-**R**ank **A**daptation trains low-rank adapters while freezing the rest of the model [Hu et al., 2022a]. We use LoRA to reduce the number of trainable parameters to 3.4M (1.37% from 250M). Using 2 clients per round, we reach the target ANLS in 7 rounds (**0.38 GB** total communication).

**2. Tuning FL hyperparameters.** On top of **1. LoRA**, we sample 1 client per round (default: 2) and train for 16 local epochs (default: 1), which respectively reduces communication and improves utility. With these adjustments, we reach the target ANLS in 2 rounds (**55 MB** total communication).

**3. Quantization** is a lossy compression approach which we use to reduce the size of the communicated LoRA updates. We use NF4 (4-bit) quantization which reduces the message size by $\sim 8\times$ while achieving the target ANLS with the same configuration as **2**. (**7.7 MB** total communication).

## 4.4 Runners-up Track 1: Niwa, Ishii, Yamasaki, Fukami, Tyou, and Yokota

We briefly present our methods and experimental results. For more detailed information can be found in Appendix D. We aimed to achieve faster convergence of training for local models with fewer communication rounds. To achieve this, we utilized Shampoo [Gupta et al., 2018], a second-order optimization method, in local update rules by multiplying the local preconditioning matrix to the local stochastic gradient. The update rules of our method, named *FedShampoo*, are outlined in Alg. 1 in Appendix D.1. Shampoo enables smooth local updates by geometrically rotating and scaling stochastic gradients. To reduce the memory footprint in computing large-scale preconditioning matrices, we approximated them by employing layer-wise block-diagonalization. Notably, the local preconditioning matrices (approximated by sub-matrices) were not transmitted to the central server, thus avoiding excess communication costs. Furthermore, we excluded the embedding layer from the optimization target, resulting in a reduction of approximately 26 % in communication per round compared to whole parameters[3].

In Table 2, FedShampoo achieved the target ANLS score with 10.01 GB communication cost. Refer to Figure A.3 in Appendix D.1 for convergence curves using validation loss, ACC and ANLS. We submitted the model after only $R = 3$ communication rounds, surpassing the target ANLS score of 0.8873 and resulting in an approximately 30 % reduction of the communication cost compared with the baseline method (using solely AdamW-based optimizer). Furthermore, FedShampoo achieved higher ACC and ANLS scores compared with the baseline method after exceeding the ANLS target

---

[3]We submitted a model applying LoRA to FedShampoo; however, it did not exceed the target ANLS score.

score (after 3 communication rounds). This provides as empirical evidence of FedShampoo's faster convergence, which benefits from applying the preconditioning matrix to the stochastic gradient. The detailed experimental configurations, such as hyperparameter tunings of learning rate and clipping threshold, are summarized in Appendix D.1.

# 5 Track 2: Differentially Private Federated Learning

Track 2 focuses on training as high utility models as possible while preserving all information from each document provider in the training set through DP. We describe the task, the organizer's baseline and two submitted approaches (See Table 3).

## 5.1 Track 2 Task Formulation

The objective of track 2 is to achieve the best utility possible while protecting all information from each document provider in the training set, which could be exposed through textual (provider company name) or visual (logo, presentation) information. Participants are required to train under DP at different levels from medium DP ($\epsilon = 1$) to weak DP ($\epsilon = 8$) to mitigate the risk of provider information being leaked. Ultimately, the goal is to achieve the best utility while complying to the privacy budgets of $\epsilon \in \{1, 4, 8\}$ at $\delta = 10^{-5}$. The definition of DP critically depends on the concept of adjacency of datasets. We seek to protect the privacy of providers and thus the typical document-level adjacency definition would be too weak, as there are many documents from the same provider and combining them could leak private information. Instead we use *provider-level add/remove adjacency*, where adjacent training datasets can be obtained by adding or removing all documents from one provider. Prior work denotes this as group-level DP [Marathe and Kanani, 2022, Galli et al., 2023].

Participants are required to follow the same rules regarding the pre-trained model and fine-tuning data as in track 1. Besides uploading the final model checkpoint solutions, they are required to submit a theoretical privacy proof and description. The requirement for a theoretical privacy proof in track 2 ensures that the solutions proposed by participants are rigorously validated for their adherence to differential privacy principles. This proof demonstrates that the final model maintains the privacy of all information from each document provider by offering a quantifiable measure of privacy loss. Additionally, a thorough description and code submission are necessary to facilitate reproducibility and allow for independent verification of the privacy claims, ensuring transparency and trustworthiness in the solutions provided.

## 5.2 Baseline Solution Track 2

The baseline solution for track 2 utilizes DP stochastic optimization. The optimization of the model is done in multiple global rounds. In each round, the central server first samples a set of clients from all $N = 10$ clients. Each selected client runs a local instance of federated learning where each provider acts as the training data of a *virtual client* within the real client. The client randomly selects providers, clips the per-provider updates and the adds an appropriate amount of noise so that the update aggregated by the server is differentially private with respect to all providers over all clients[4] The privacy loss of the baseline follows the usual analysis of DP stochastic optimisation consisting of compositions of sub-sampled Gaussian mechanisms. The loss depends on the number of iterations $T_{cl}$, sub-sampling rate $q$ (both over clients and providers) and noise scale $\sigma$ [Mironov et al., 2019, Balle et al., 2020]. (See more details in Appendix A.4 and the privacy analysis in Appendix B).

## 5.3 Winner Track 2: Ragul N and Kutum

Similar to the winning solution for track 1, our method also uses LoRA. We choose LoRA for the following two reasons: First, it significantly reduces the communication cost as shown in Section 4.3.

---

[4]Note when no clients are sampled in a FL round the server still needs to add noise.

Table 3: Competition Winners Track 2 (Differential Private Federated Learning)

| Rank | Team | Method | ANLS ↑ | | |
|---|---|---|---|---|---|
| | | | at $\epsilon = 1$ | at $\epsilon = 4$ | at $\epsilon = 8$ |
| 1 | Ragul N and Kutum (Section 5.3) | LoRA | 0.5854 | 0.6121 | 0.6225 |
| 2 | Fukami et al. (Section 5.4) | DP-CLGECL | 0.5724 | 0.6018 | 0.6033 |
| - | Organizers (Section 5.2) | Baseline | 0.4832 | 0.5024 | 0.5132 |

Second, empirical results have shown that differentially private adaptation of language models using parameter-efficient methods such as LoRA outperforms full fine-tuning in centralized settings [Yu et al., 2022b]. These methods reduce the overall noise added by only updating a small proportion of the parameters in the model, thereby increasing the utility of the model. The communication efficiency of LoRA also allowed us to increase the number of FL rounds from 5 in the baseline method to 30 in our method without increasing communication costs. With these changes to the baseline, our method improved the ANLS by 10-11 percentage points across all privacy settings.

## 5.4 Runners-up Track 2: Fukami, Yamasaki, Niwa, and Tyou

We briefly present our methods and experimental results. More detailed information can be found in Appendix D. It is well-known that applying DP to FedAVG with a relatively high privacy level often stagnates the model training process due to local parameter drift. This is mainly caused by i) noise addition in DP and ii) data heterogeneity among clients. To address these issues, we propose *DP-CLGECL*, which incorporates the DP's Gaussian mechanism into CLGECL Tyou et al. [2024]. The update rules in DP-CLGECL are derived by solving a linearly constrained loss-sum minimization problem, resulting in robustness against local gradient drift due to data heterogeneity, and this would also be effective in addressing the drift issue due to DP's Gaussian mechanism. Note that the DP analysis of the private baseline detailed in Appendix B is applicable to our DP-CLGECL. More details about our methodologies are provided in Appendix D.2.

As indicated in Table 3, ANLS showed significant improvement with the use of our DP-CLGECL compared with the baseline method for each $\varepsilon$. Associated experimental results, including convergence curves in Figure A.4 are summarized in Appendix D.2. After passing the competition deadline, we observed a negative impact of using AdamW optimizer in the baseline method. The norm of stochastic gradient, preconditioned by AdamW, often increased, and the gradient clipping used to ensure the pre-defined DP levels led to a loss of valuable information in model parameter training. To address this issue, we replaced AdamW with momentum in the local update of DP-CLGECL, resulting in further improved ANLS. Although more details can be found in Figure A.5, the ANLS was then $0.5918$ for $\varepsilon = 1$ using DP-CLGECL with momentum.

## 6 Lessons Learnt and Recommendations for Future FL and DP Competitions

In this section we present lessons learnt from organizing this competition and discuss best practices that could be considered for organizing competitions in the future.

### 6.1 Ensuring that the Track 2 Submissions Are DP

The track 2 of this competition required participants to provide a model checkpoint trained under DP. Additionally, we asked the participants to provide a privacy proof outlining how their method is formally differential private and requested the source code.

**Formal privacy proof** Asking for a privacy proof from the participants results in two things: (i) The organizers can check that a new proposed method is DP; and (ii) The participating team can reflect on ensuring that their method is actually DP. Insufficient formal analysis in prior work has lead to response papers [Carlini et al., 2021, 2022] that corrected the wrong analysis.

**Ensuring that the implementations are DP** While the privacy proof ensures that theoretically the submissions are DP, even small mistakes in the implementation of DP methods can invalidate

or severely weaken the DP guarantees [Tramèr et al., 2022b]. Among these are the clipping of the updates, the correct noise addition and scaling as well as the subsampling. Thus, members of the organizing team have inspected the implementations of the best scoring methods but this is a manual process that does not scale to competitions with a large number of participants. The code reviews could be complemented with automatic tests that increase the chance of finding bugs in the implementation. Established DP libraries such as Opacas [Yousefpour et al., 2021] use unit tests but these tests are custom to the implementation that are testing and writing new tests requires much more manual labour than plain code reviews. Using only established implementations (e.g., like Opacus) for critical parts of the code would reduce the risk of bugs but also limit the possible solutions.

**Automation of the validation of DP methods and implementations** When scaling up the participant numbers of a competition, processes need to be automated. One example for that is our automatic utility evaluation on the secret test set. Automating the validation of DP methods and implementations is less straightforward: There are methods for auditing DP implementations [Jagielski et al., 2020, Nasr et al., 2023] but they are computationally expensive. Recent advancements have significantly reduced the cost of DP auditing [Steinke et al., 2023]. One option would be auditing new submissions to assist in DP validation but it is unclear how computationally costly that would be. Auditing cannot conclusively prove something DP, so it should only be used to complement privacy proofs and code checks, not replace them.

### 6.2 Lowering the Threshold for Participation

Referring to Table 1 one can see that the competition has received some interest. Also, it led to the data set being adopted in the privacy community [Wu et al., 2024] and increased the awareness in the document intelligence community [Biescas et al., 2024]. Participants were required to be able to train a state-of-the-art Document Visual Question Answering model in a federated learning setting (under DP). The number of potential participants that have the required skill set is not as high as in other challenges. Thus it is important that the threshold for participation is as low as possible. We discuss measures that we took to lower the threshold for participation.

**Starting Kit** All solutions that are described in this analysis report utilized the provided starting kit to some extent. Based on the feedback from the participants, we think that the starting kit was crucial for them to participate. We can recommend to future organizers to test and document the starting kit extensively and include convenience functions (e.g., to compute communication cost or DP noise).

**Computational Cost** Simulating the FL setting and even just fine-tuning large pre-trained models requires a significant amount of compute. This is especially true under DP as the privacy/utility trade-off can be improved by training longer [Ponomareva et al., 2023] and using larger batch sizes [Räisä et al., 2024]. We aimed to lower the threshold for participation by reducing the size of the client datasets and utilizing not the largest pre-trained model available. Still, executing the baselines with consumer hardware is hard if not impossible. One possible avenue for the future would be to open separate tracks for consumer hardware and provide cloud compute to teams that could otherwise not participate. The recent NeurIPS 2023 challenge on LLMs[5] introduced some of these measures.

## 7 Conclusion & Outlook

The challenge is a benchmark and remains open for future submissions. In the future, we will host a red team challenge, where teams run privacy attacks against models from this challenge.

**Broader Impact** This challenge invited the community to design novel creative solutions for real-life use cases. This has significant positive impact on users training ML models on personal data. The best practices and our setup can be used to improve further challenges.

**Limitations** This challenge only focused on training models but does not focus on other parts of machine learning systems that may be vulnerable to privacy attacks as well [Debenedetti et al., 2023].

---

[5]LLM Efficiency Challenge: 1LLM+1GPU+1Day:https://llm-efficiency-challenge.github.io/

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

# A  General Appendix

## A.1  Author contributions

In this section we list the author contributions. The participants wrote the Sections 4.3, 4.4, 5.3 and 5.4.

**Organizers of the challenge**:

Marlon Tobaben[1], Mohamed Ali Souibgui[2], Rubèn Tito[2], Khanh Nguyen[2], Raouf Kerkouche[3], Kangsoo Jung[4], Joonas Jälkö[1], Lei Kang[2], Andrey Barsky[2], Vincent Poulain d'Andecy[5], Aurélie JOSEPH[5], Josep Llados[2], Ernest Valveny[2], Antti Honkela[1], Mario Fritz[3], Dimosthenis Karatzas[2]

[1]University of Helsinki, [2]Computer Vision Center, Universitat Autònoma de Barcelona, [3]CISPA Helmholtz Center for Information Security, [4]INRIA, [5]Yooz

**Winners Track 1:**

Aashiq Muhamed[6], Kevin Kuo[6], Virginia Smith[6]

[6]Carnegie Mellon University

**Track 1 runners-up:**

Kenta Niwa[7], Hiro Ishii[8], Yusuke Yamasaki[7], Takumi Fukami[7], Iifan Tyou[7], Rio Yokota[8]

[7]NTT, [8]Tokyo Institute of Technology

**Winners Track 2:**

Ragul N[9], Rintu Kutum[9]

[9]Asoka University

**Track 2 runners-up:**

Takumi Fukami[7] , Yusuke Yamasaki[7] , Kenta Niwa[7] , Iifan Tyou[7]

[7]NTT

## A.2 Dataset

This section contains additional information regarding the dataset. The data set is described in more detail in Tito et al. [2023b] and is available to download on the ELSA benchmark platform `https://benchmarks.elsa-ai.eu/?ch=2&com=downloads`. The Dataset is created using images from the DocILE dataset [Šimsa et al., 2023], which was published under the MIT License. For PFL-DocVQA we created new annotations for these images. The created annotations are the OCR transcriptions (using Amazon Textract) and the pairs of question/answer. The question/answer pairs are generated using key/value pairs extracted by Amazon Textract and then manually verified. For each key, a question is formed to ask about it, and the answer is the corresponding value. These questions are generated semi-automatically by creating multiple templates for each key and then using a language model OpenAI [2023] to rephrase them, achieving linguistic diversity. Our dataset is published under the Licence CC-BY-4.0.

| Dataset | Client (Subset) | Provider | Document | Page | Question/Answer |
|---|---|---|---|---|---|
| Train | 0 | 400 | 2224 | 5930 | 19465 |
| | 1 | 418 | 2382 | 6694 | 22229 |
| | 2 | 404 | 2296 | 6667 | 21673 |
| | 3 | 414 | 2358 | 6751 | 22148 |
| | 4 | 429 | 4543 | 12071 | 32472 |
| | 5 | 423 | 2378 | 6984 | 22361 |
| | 6 | 423 | 2700 | 7406 | 23801 |
| | 7 | 416 | 1951 | 5617 | 18462 |
| | 8 | 401 | 1932 | 5421 | 17868 |
| | 9 | 421 | 2136 | 6353 | 20840 |
| Valid | - | 2231 | 3536 | 9150 | 30491 |
| Test | In-Distribution | 1390 | 2875 | 8088 | 25603 |
| | Out-of-Distribution | 977 | 1912 | 5375 | 17988 |

Table A1: Statistics on the base PFL-DocVQA Dataset in terms of number of Providers/Documents/Pages/Question-Answers.

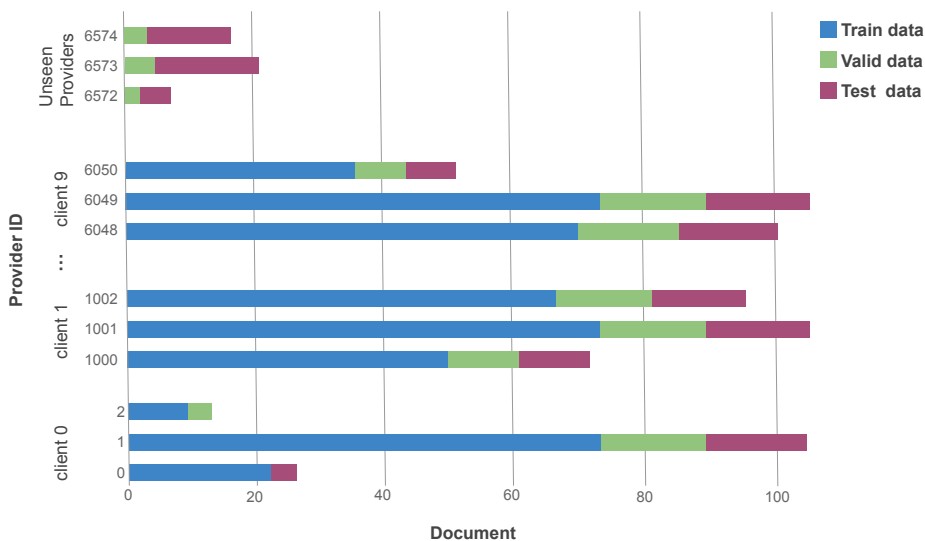

Figure A.1: Data split of the PFL-DocVQA dataset.

## A.3 Additional information on the model

The pre-trained model [Tito et al., 2023a] can be found at `https://huggingface.co/rubentito/` `vt5-base-spdocvqa`. It is licensed under the gpl-3.0 license.

## A.4 Training details for baselines

The hyperparameters for the baseline were chosen using a combination of grid search and manual search. The assumption for the baselines is not to have optimal hyperparameters but rather reasonable baselines.

We utilize two NVIDIA A40 (40 GB VRAM each) and train for some hours to obtain the baselines. The exact runtime depends on the hyperparamters being used.

### A.4.1 Track 1

This baseline achieves 0.8676 of ANLS and 77.41 accuracy on the validation set after 10 FL Rounds. It transmits 1.12GB constantly for each communication stream, which results in a total of 44.66GB during the entire training process. We sample $K = 2$ clients at every federated round.

### A.4.2 Track 2

The baseline is obtained through 5 FL Rounds. It transmits 1.12GB constantly for each communication stream, which results in a total of 22.32GB during the entire training process. We sample $K = 2$ clients per round and $M = 50$ providers on each client. The updates are clipped to a norm of 0.5 and the Gaussian noise is computed so that the privacy budgets of $\epsilon \in \{1, 4, 8\}$ at $\delta = 10^{-5}$.

## B Privacy Analysis

The privacy analysis of our differentially private baseline is discussed in this section. The provided python script to compute the privacy budget $\varepsilon$ is derived from the following analysis.

### B.1 Definitions

**Definition B.1** (Differential Privacy Dwork and Roth [2014]). A randomized mechanism $\mathcal{M}$ with range $\mathcal{R}$ satisfies $(\varepsilon, \delta)$-differential privacy, if for any two adjacent datasets $E$ and $E'$, i.e., $E' =$

$E \cup \{x\}$ for some $x$ in the data domain (or vice versa), and for any subset of outputs $O \subseteq \mathcal{R}$, it holds that

$$\Pr[\mathcal{M}(E) \in O] \le e^{\varepsilon} \Pr[\mathcal{M}(E') \in O] + \delta \tag{A1}$$

Intuitively, DP guarantees that an adversary, provided with the output of $\mathcal{M}$, can draw almost the same conclusions (up to $\varepsilon$ with probability larger than $1 - \delta$) about any group no matter if it is included in the input of $\mathcal{M}$ or not Dwork and Roth [2014]. This means, for any group owner, a privacy breach is unlikely to be due to its participation in the dataset.

In Federated Learning, the notion of *adjacent (neighboring) datasets* used in DP generally refers to pairs of datasets differing by one client (*client-level* DP), or by one group of one user (*group-level* DP), or by one data point of one user (*record-level* DP). Our challenge focuses on the *group-level* DP Galli et al. [2023], where each group refers to a provider.

We use the Gaussian mechanism to upper bound privacy leakage when transmitting information from clients to the server.

**Definition B.2.** (Gaussian Mechanism Dwork and Roth [2014]) Let $f : \mathbb{R}^n \to \mathbb{R}^d$ be an arbitrary function that maps $n$-dimensional input to $d$ logits with sensitivity being:

$$S = \max_{E, E'} \|f(E) - f(E')\|_2 \tag{A2}$$

over all adjacent datasets $E$ and $E' \in \mathcal{E}$. The Gaussian Mechanism $\mathcal{M}_{\sigma}$, parameterized by $\sigma$, adds noise into the output, i.e.,

$$\mathcal{M}_{\sigma}(x) = f(x) + \mathcal{N}(0, \sigma^2 I). \tag{A3}$$

As in Abadi et al. [2016], Mironov et al. [2019], we consider the Sampled Gaussian Mechanism (SGM) —a composition of subsampling and the additive Gaussian noise (defined in B.5)— for privacy amplification. Moreover, we first compute the SGM's Renyi Differential Privacy as in Mironov et al. [2019] and then we use conversion Theorem B.8 from Balle et al. [2020] for switching back to Differential Privacy.

**Definition B.3** (Rényi divergence). Let $P$ and $Q$ two distributions on $\mathcal{X}$ defined over the same probability space, and let $p$ and $q$ be their respective densities. The Rényi divergence of a finite order $\alpha \ne 1$ between $P$ and $Q$ is defined as follows:

$$D_{\alpha}(P \parallel Q) \triangleq \frac{1}{\alpha - 1} \ln \int_{\mathcal{X}} q(x) \left( \frac{p(x)}{q(x)} \right)^{\alpha} dx.$$

Rényi divergence at orders $\alpha = 1, \infty$ are defined by continuity.

**Definition B.4** (Rényi differential privacy (RDP)). A randomized mechanism $\mathcal{M} : \mathcal{E} \to \mathcal{R}$ satisfies $(\alpha, \rho)$-Rényi differential privacy (RDP) if for any two adjacent inputs $E, E' \in \mathcal{E}$ it holds that

$$D_{\alpha}(\mathcal{M}(E) \parallel \mathcal{M}(E')) \le \rho$$

In this work, we call two datasets $E, E'$ to be adjacent if $E' = E \cup \{x\}$ (or vice versa).

**Definition B.5** (Sampled Gaussian Mechanism (SGM)). Let $f$ be an arbitrary function mapping subsets of $\mathcal{E}$ to $\mathbb{R}^d$. We define the Sampled Gaussian mechanism (SGM) parametrized with the sampling rate $0 < q \le 1$ and the noise $\sigma > 0$ as

$$\mathrm{SG}_{q,\sigma} \triangleq f\left(\{x : x \in E \text{ is sampled with probability } q\}\right) + \mathcal{N}(0, \sigma^2 \mathbb{I}^d),$$

where each element of $E$ is independently and randomly sampled with probability $q$ without replacement.

As for the Gaussian Mechanism, the sampled Gaussian mechanism consists of adding i.i.d Gaussian noise with zero mean and variance $\sigma^2$ to each coordinate value of the true output of $f$. In fact, the sampled Gaussian mechanism draws vector values from a multivariate spherical (or isotropic) Gaussian distribution which is described by random variable $\mathcal{N}(0, \sigma^2\mathbb{I}^d)$, where $d$ is omitted if it is unambiguous in the given context.

## B.2 Analysis

The privacy guarantee of FL-GROUP-DP is quantified using the revisited moment accountant Mironov et al. [2019] that restates the moments accountant introduced in Abadi et al. [2016] using the notion of Rényi differential privacy (RDP) defined in Mironov [2017].

Let $\mu_0$ denote the $\mathrm{pdf}$ of $\mathcal{N}(0, \sigma^2)$ and let $\mu_1$ denote the $\mathrm{pdf}$ of $\mathcal{N}(1, \sigma^2)$. Let $\mu$ be the mixture of two Gaussians $\mu = (1 - q)\mu_0 + q\mu_1$, where $q$ is the sampling probability of a single record in a single round.

**Theorem B.6.** *Mironov et al. [2019]. Let* $\mathrm{SG}_{q,\sigma}$ *be the Sampled Gaussian mechanism for some function $f$ and under the assumption $\Delta_2 f \leq 1$ for any adjacent $E, E' \in \mathcal{E}$. Then* $\mathrm{SG}_{q,\sigma}$ *satisfies* $(\alpha, \rho)$-*RDP if*

$$\rho \leq \frac{1}{\alpha - 1} \log \max(A_\alpha, B_\alpha) \tag{A4}$$

*where* $A_\alpha \overset{\Delta}{=} \mathbb{E}_{z \sim \mu_0}[(\mu(z)/\mu_0(z))^\alpha]$ *and* $B_\alpha \overset{\Delta}{=} \mathbb{E}_{z \sim \mu}[(\mu_0(z)/\mu(z))^\alpha]$

Theorem B.6 states that applying SGM to a function of sensitivity (Equation B.2) at most 1 (which also holds for larger values without loss of generality) satisfies $(\alpha, \rho)$-RDP if $\rho \leq \frac{1}{\alpha-1} \log(\max\{A_\alpha, B_\alpha\})$. Thus, analyzing RDP properties of SGM is equivalent to upper bounding $A_\alpha$ and $B_\alpha$.

From Corollary 7. in Mironov et al. [2019], $A_\alpha \geq B_\alpha$ for any $\alpha \geq 1$. Therefore, we can reformulate A4 as

$$\rho \leq \xi_\mathcal{N}(\alpha|q) := \frac{1}{\alpha - 1} \log A_\alpha \tag{A5}$$

To compute $A_\alpha$, we use the numerically stable computation approach proposed in Mironov et al. [2019] (Sec. 3.3) depending on whether $\alpha$ is expressed as an integer or a real value.

**Theorem B.7** (Composability Mironov [2017]). *Suppose that a mechanism $\mathcal{M}$ consists of a sequence of adaptive mechanisms $\mathcal{M}_1, \ldots, \mathcal{M}_k$ where $\mathcal{M}_i : \prod_{j=1}^{i-1} \mathcal{R}_j \times \mathcal{E} \to \mathcal{R}_i$. If all the mechanisms in the sequence are $(\alpha, \rho)$-RDP, then the composition of the sequence is $(\alpha, k\rho)$-RDP.*

In particular, Theorem B.7 holds when the mechanisms themselves are chosen based on the (public) output of the previous mechanisms. By Theorem B.7, it suffices to compute $\xi_\mathcal{N}(\alpha|q)$ at each step and sum them up to bound the overall RDP privacy budget of an iterative mechanism composed of single DP mechanisms at each step.

**Theorem B.8** (Conversion from RDP to DP Balle et al. [2020]). *If a mechanism $\mathcal{M}$ is $(\alpha, \rho)$-RDP then it is $((\rho + \log((\alpha - 1)/\alpha) - (\log \delta + \log \alpha)/(\alpha - 1), \delta)$-DP for any $0 < \delta < 1$.*

**Theorem B.9** (Privacy of FL-GROUP-DP). *For any $0 < \delta < 1$ and $\alpha \geq 1$, FL-GROUP-DP is $(\min_\alpha(T_{\mathrm{cl}} \cdot \xi(\alpha|q) + \log((\alpha - 1)/\alpha) - (\log \delta + \log \alpha)/(\alpha - 1)), \delta)$-DP, where $\xi_\mathcal{N}(\alpha|q)$ is defined in Eq. A5, $q = \frac{C \cdot |\mathbb{M}|}{\min_k |\mathbb{G}_k|}$.*

The proof follows from Theorems B.6, B.7,B.8 and the fact that a group (provider) is sampled in every federated round if (1) the corresponding client is sampled, which has a probability of $C$, and (2) the batch of groups sampled locally at this client contains the group, which has a probability of at most $\frac{|\mathbb{M}|}{\min_k |\mathbb{G}_k|}$. Therefore, a group is sampled with a probability of $q = \frac{C \cdot |\mathbb{M}|}{\min_k |\mathbb{G}_k|}$.

# C   Supplementary Information of Section 4.3

Here, we present details for reproducing the results from Section 4.3. In all experiments, clients perform local fine-tuning with batch size = 16 and learning rate = 2e-4. In our code, we train one model at a time using data parallelism. Specifically, we split each batch over 8 GPUs, resulting in a batch size of 2 per GPU (we used 8 GeForce GTX 1080 Ti GPUs). Our code will be shared on Github: https://github.com/imkevinkuo/PFL-DocVQA-Competition.

## C.1   Communication cost

Since all messages have an identical size in this FL setting, the total communication cost is simply a product of the a) size of communicated messages and b) number of messages sent. In the table below, we break down each method's cost using the following equations:

$$
\begin{aligned}
\text{`Total'} &= \text{`Message Size'} & \times \ \text{`Messages'} \\
\text{where `Message Size'} &= \big(\text{`LoRA' (\#params)} & + \ \text{`Base (\#params)'}\big) & \times \ \text{`Bits' (per param)} \\
\text{and `Messages'} &= \text{`C' (clients per round)} & \times \ \text{`R' (FL rounds)} & \times \ 2 \ (\text{up and down})
\end{aligned}
$$

| | Message Size | | | | Messages | | Total | ANLS | |
| --- | --- | --- | --- | --- | --- | --- | --- | --- | --- |
| Method | LoRA | Base | Bits | Bytes | C | R | Bytes | Val | Test |
| Baseline | - | 250M | 32 | 1.11 GB | 2 | 10 | 40 GB | .8676 | .8873 |
| LoRA (rank=6) | 660K | 2.75M | 32 | 13.7 MB | 2 | 7 | 380 MB | .8400 | .8566 |
| Tuned HPs | 660K | 2.75M | 32 | 13.7 MB | 1 | 2 | 55 MB | .8467 | .8683 |
| Quantization | 660K | 2.75M | 4.5 | 1.92 MB | 1 | 2 | 7.7 MB | .8444 | .8673 |

Table A2: We summarize the three methods used. LoRA reduces the number of trainable parameters, tuning HPs reduces the number of messages, and quantization reduces the parameter bitwidth.

**LoRA.** While the VT5 architecture contains both a language backbone (T5) and vision backbone (DiT), we only use LoRA on the language backbone and insert 110K new parameters per LoRA rank. For the vision backbone ('Base'), we directly fine-tune the spatial encoder (2.16M params) and visual embedding projection head (0.59M params). All other parameters in the entire model are frozen. Although LoRA changes the model architecture during training, it can be merged with the pretrained architecture after training is complete, which allowed us to make valid submissions.

The $\sim$ 110K parameters (0.44 MB) per LoRA rank $r$ come from applying LoRA to the query and value projections in each attention layer of the language backbone. Each projection matrix has dimension $768 \times 768$, so its two adapter matrices $A, B$ will both have dimension $768 \times r$. There are 36 attention layers which contain a query and value projection, giving the final value:

$$
36 \text{ (layers)} \ \times \ 2 \text{ (query and value)} \times \ 2 \text{ (A and B)} \ \times \ 768 \ \times \ r \text{ (rank)} \ = \ 110,592 \approx 110\text{K} \times r
$$

We note that LoRA typically takes more iterations to train than full fine-tuning. While the full fine-tuning baseline provided by the organizers achieves **.8242** validation ANLS in 4 rounds (this is 5% below the .8676 ANLS at 10 rounds), we find that LoRA takes 7 rounds ($\uparrow 2\times$) to achieve the same ANLS. However, the parameter reduction from LoRA ($\downarrow 100\times$) greatly offsets this cost. For all experiments in this section, we use LoRA with rank $r = 6$.

## C.2   Tuned FL Hyperparameters

We find that extended local fine-tuning on a single client is very helpful, as it increases utility with no additional communication cost. In Table A3, we show that training only on a single client can achieve .8242 ANLS. We also find that sampling a single client is more efficient than averaging multiple clients each round. In Table A4, '1 Client' usually outperforms '2 Clients' when given double the number of rounds.

| Epochs | Client ID 0 | 1 | 2 | 9 |
|---|---|---|---|---|
| 1 | .7648 | .7638 | .7577 | .7552 |
| 2 | .7893 | .7912 | .7904 | .7797 |
| 4 | .8111 | .8108 | .8039 | .8089 |
| 8 | .8247 | .8219 | .8231 | .8176 |
| 16 | .8337 | .8345 | .8329 | .8307 |

Table A3: Extended local training on a single client greatly improves validation ANLS.

| 1 Client | FL Rounds 1 | 2 | 4 | 8 |
|---|---|---|---|---|
| 1 Epoch | .7419 | .7875 | .8083 | .8331 |
| 2 Epochs | .7719 | .8061 | .8206 | .8382 |
| 2 Clients | (2× communication cost) | | | |
| 1 Epoch | .7493 | .7899 | .8232 | .8400 |
| 2 Epochs | .7696 | .8083 | .8355 | .8513 |

Table A4: Sampling one client and training for double the rounds achieves a higher validation ANLS than sampling two clients.

One surprising takeaway from our experiments is that the data from a single client is adequate to train a competitive model. However, there are many limitations with limiting the client subsample, which we briefly outline. First, in cross-device FL settings which consider a large network (up to millions) of clients, extreme subsampling can lead to low-quality global updates. Next, since subsampling slows down convergence, the model will take more rounds and thus more wall-clock time to train. Finally, in the context of privacy, sampling fewer clients makes it more difficult to bound the sensitivity of the aggregate update with respect to any individual client's data, which results in greater privacy loss.

### C.3 Quantization

By default, each parameter is communicated as a 32-bit floating-point value (FP32). We reduce this to 4.5 bits ($\downarrow 7\times$) by using **NF4** (normal-float) quantization [Dettmers et al., 2023]. While NF4 proposes using LoRA on top of a quantized backbone, we use quantization to reduce the size of all communicated parameters (in both LoRA and the backbone). Similar recent FL methods have generally explored combining LoRA with parameter compression to reduce communication [Yadav et al., 2023, Kuo et al., 2024].

In NF4, each parameter is stored using 4 bits (16 unique values) and each block of $k = 64$ parameters shares an FP32 normalization factor. This adds up to $4 + (32/k) = 4.5$ bits per parameter, as shown in Table A2. Parameters are quantized **only before communication**, while finetuning and aggregation are all done in full precision. As we show in Table A5, quantization slightly harms model performance, but this cost is greatly offset by the reduction in communication.

| Round | Stage | Full-precision 1 Client | 2 Clients | Quantized 1 Client | 2 Clients |
|---|---|---|---|---|---|
| 1 | Download | Initialize weights using shared RNG seed | | | |
|  | Finetuning | .8337 | .8341 | .8337 | .8341 |
|  | Upload | - | - | .8301 | .8313 |
|  | Aggregation | - | .8255 | - | .8253 |
| 2 | Download | - | - | - | .8253 |
|  | Finetuning | **.8467** | .8437 | .8448 | 8445 |
|  | Upload | - | - | **.8444** | .8524 |
|  | Aggregation | - | **.8520** | - | **.8518** |
| Total Communication | | 55 MB | 110 MB | 7.7 MB | 15.4 MB |

Table A5: We track the validation ANLS after each stage of communication-efficient FL. When sampling '2 Clients' per round, 'Finetuning' and 'Upload' refer to the average ANLS over the two client models. '-' indicates that the same model(s) are evaluated as the cell above e.g. full-precision 'Upload' and 'Download' do not change the model(s).

# D  Supplementary Information of Sections 4.4 and 5.4

## D.1  FedShampoo for Track 1

**Update rules of FedShampoo**: First, we explain the update rule using Shampoo Gupta et al. [2018]. As discussed in Sec. 4.4, Shampoo is a second-order optimization method that involves multiplying the preconditioning matrix with the (stochastic) gradient, and the preconditioning technique in Shampoo is introduced in the local model update in our FedShampoo, which is summarized in Alg. 1.

In the optimization of models in the form of neural networks, it is typical for model parameters to be described by a stack of matrices/tensors to transform each layer's input and output. Although we have focused on formulating the update rules in a matrix manner (since we will mainly focus on Transformer-based model), it is not a loss of generality. For all clients $i \in [N]$ and each layer $b \in [B]$, let $W_{i,b}^{(t)} \in \mathbb{R}^{d_{\text{out},b} \times d_{\text{in},b}}$ be the model parameter in the $b$-th layer of the neural network, and $G_{i,b}^{(t)} \in \mathbb{R}^{d_{\text{out},b} \times d_{\text{in},b}}$ be the stochastic gradient of the local loss function with respect to $W_{i,b}^{(t)}$. The local model update rule using Shampoo is given by

$$L_{i,b}^{(t+1)} = L_{i,b}^{(t)} + G_{i,b}^{(t)} \left[ G_{i,b}^{(t)} \right]^{\top},$$

$$R_{i,b}^{(t+1)} = R_{i,b}^{(t)} + \left[ G_{i,b}^{(t)} \right]^{\top} G_{i,b}^{(t)},$$

$$W_{i,b}^{(t+1)} = W_{i,b}^{(t)} - \eta \left[ L_{i,b}^{(t)} \right]^{-1/4} G_{i,b}^{(t)} \left[ R_{i,b}^{(t)} \right]^{-1/4}, \tag{A6}$$

where $\eta$ denotes the learning rate, and $L_i^{(t)} \in \mathbb{R}^{d_{\text{out},b} \times d_{\text{out},b}}$ and $R_i^{(t)} \in \mathbb{R}^{d_{\text{in},b} \times d_{\text{in},b}}$ are the preconditioning matrices for the gradient and the weight matrix, respectively.

In Eq. equation A6, the local preconditioning matrices, $L_{i,b}$ and $R_{i,b}$, are multiplied to both sides of the stochastic gradient in a matrix form $G_{i,b}$. This process can be interpreted as mitigating changes in the local gradient of loss function through model parameter updates by multiplying local preconditioning matrices. This supports mitigating the negative effects of complex loss landscape in the loss function using neural networks, and it can lead to fast convergence to the stationary point.

Thanks to the Shampoo application in a layer-wise manner, it is possible to track $L_{i,b}$ and $R_{i,b}$ for each layer, which significantly reduces the memory footprint. Specifically, while the full-matrix version of AdaGrad Duchi et al. [2010] requires memory linearly proportional to the number of model parameters $O(d_{\text{out},b}^2 d_{\text{in},b}^2)$, Shampoo only requires memory with $O(d_{\text{out},b}^2 + d_{\text{in},b}^2)$ for each layer. Furthermore, the inversion of the preconditioning matrices can be efficient, since it takes $O(d_{\text{out},b}^3 + d_{\text{in},b}^3)$ rather than $O(d_{\text{out},b}^3 d_{\text{in},b}^3)$ in terms of computational complexity.

Additionally, element-wise clipping was used in the local model update rule, which is a de-facto standard for stable optimization of the Transformer-based models, as mentioned in e.g., Zhang et al. [2020]. Due to the heavy-tailed noise in stochastic gradient, the magnitude of updates in model parameters has significantly changed, leading to unstable convergence. To address this issue, we effectively alleviated this phenomenon by incorporating the clipping of the magnitude of each element of gradients into adaptive updates using Shampoo.

Finally, as noted in Sec. 4.4, to reduce the amount of communication per round, the embedding layer was excluded from the optimization target. This results in a reduction of around 26 % amount of parameters, rather than transmitting whole parameters.

In the following, experimental setups are explained.

**Compared methods**: In our experiment, we utilized two methods with differing local update rules: 1) the baseline method using AdamW optimizer, and 2) FedShampoo using Shampoo-based preconditioner to the Stochastic Gradient Descent (SGD).

**Hyperparameter Tuning**: To ensure a fair comparison of the two methods, several hyperparameters (learning rate $\eta$ and element-wise clipping threshold $C$) were empirically tuned. This was done while

---

**Algorithm 1** Update rules of FedShampoo

---

1: ▷ Initialization $w_i, L_{i,b} = I, R_{i,b} = I, \rho_L = \rho_R = 1e^{-4}$

2: **for** $r \in \{1, \dots, R\}$ (Outer loop round) **do**
3:    ▷ (i) Global model update in central server
4:    ▷ Averaging of aggregated local models
     $\bar{w} = \frac{1}{K} \sum_{i=1}^{K} w_i$
5:    ▷ Transmit global model to clients
     **Transmit**$_{\text{server}\rightarrow\text{client}}(\bar{w})$

6:    ▷ (ii) Local model updates in each client
7:    **for** $i \sim [N]$ ($K = 2$ client sampling) **do**
8:       ▷ Initialization of local model
       $w_i \leftarrow \bar{w}$
9:       **for** $t \in \{1, \dots, T\}$ (Inner loop iteration) **do**
10:          ▷ Local stochastic gradient $g_i \in \mathbb{R}^d$
11:          **for** $b \in \{1, \dots, B\}$ (Layer-wise iteration) **do**
12:             ▷ Reshaping elements of $g_i$ regarding $b$-th layer to be a matrix form
              $G_{i,b} \in \mathbb{R}^{d_{in,b} \times d_{out,b}}$
13:             **if** $\text{mod}(t, 10) == 0$ **then**
14:                ▷ Local update of preconditioning matrices using moving average
                $L_{i,b} \leftarrow L_{i,b} + G_{i,b}[G_{i,b}]^\top, R_{i,b} \leftarrow R_{i,b} + [G_{i,b}]^\top G_{i,b}$
15:             **end if**
16:             **if** $\text{mod}(t, 100) == 0$ **then**
17:                ▷ Computing of local preconditioning matrices
                $\tilde{L}_{i,b} \leftarrow [L_{i,b} + \rho_L I]^{-1/4}$
                $\tilde{R}_{i,b} \leftarrow [R_{i,b} + \rho_R I]^{-1/4}$
18:             **end if**
19:             ▷ Local model update using element-wise clipping
            $W_{i,b} \leftarrow W_{i,b} - \eta \, \text{Clip}(\tilde{L}_{i,b} G_{i,b} \tilde{R}_{i,b}, C)$
20:          **end for**
21:       **end for**
22:       ▷ Reshaping model in a matrix form into a vector
       $w_i \leftarrow \text{Vec}([W_{i,1}, \dots, W_{i,B}])$
23:       ▷ Transmit local model to central server
       **Transmit**$_{\text{client}_k \rightarrow \text{server}}(w_i)$
24:    **end for**
25: **end for**

---

maintaining fixed values for the total communication rounds $R = 10$, the number of inner loops for local update $L = 5000$, and the number of client sampling $K = 2$. In Fig. A.2, a summary of our hyperparameter tuning for FedShampoo is provided. After performing empirical trials, we selected $\eta = 2e^{-4}$ and $C = 0.2$.

**Computing environment**: We used a server with 8 GPUs (NVIDIA A6000 for NVLink 40GiB HBM2) and 2CPUs (Xeon).

**Experiment results**: The best validation accuracy and ANLS were achieved with the proposed FedShampoo (with freezing embedding layer). As depicted with two lines, there was a confirmed difference between the two methods.

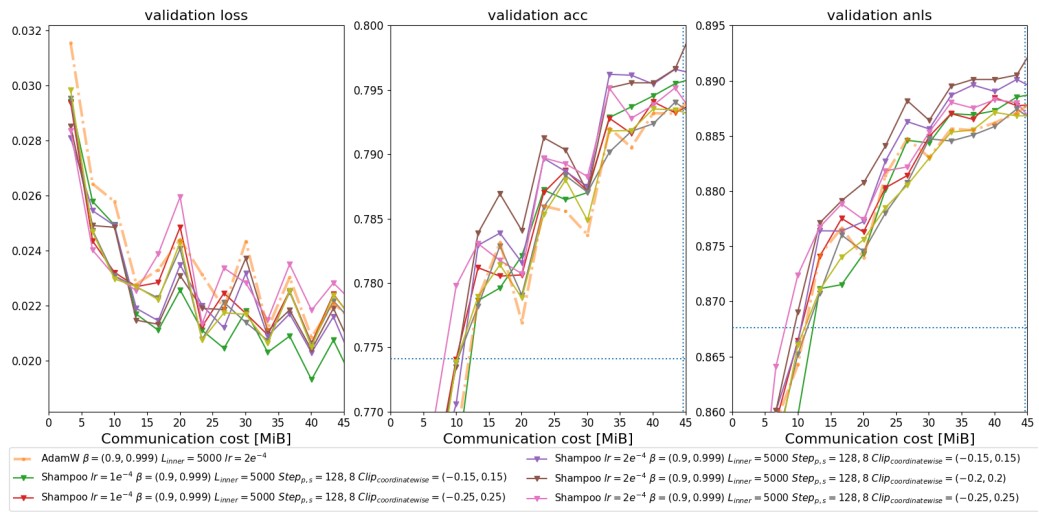

Figure A.2: Hyperparameter tuning for FedShampoo

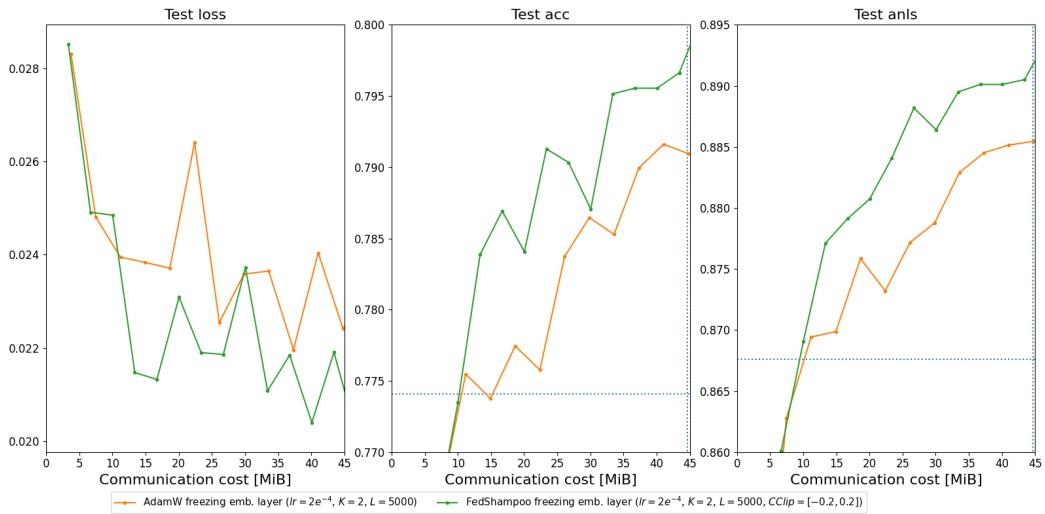

Figure A.3: Convergence curves for the global model using (Left) validation loss, (Center) validation accuracy, and (Right) validation ANLS.

 **D.2 DP-CLGECL for Track 2**

1013 Firstly, we provide a brief explanation of the formulation of CLGECL Tyou et al. [2024]. For FL
1014 consisting of $n$ local clients and a central server, we aim to solve a loss-sum minimization problem
1015 with linear constraints on local parameters $\{w^i\}_{i=1}^n$:

$$\min_{\{w^i\}_{i=1}^n} \frac{1}{n} \sum_{i=1}^n f^i(w^i) \quad \text{s.t. } w^i = w^j \quad (\forall i \in \mathbb{N}, j \in \mathbb{E}^i), \tag{A7}$$

1016 where $f^i$ represents the local loss function and $\{1, \ldots, n\} \in \mathbb{N}$, $\{1, \ldots, i-1, i+1, \ldots, n\} \in \mathbb{E}^i$.
1017 The derivation details can be found in Tyou et al. [2024]. A solver for equation A7 over the centralized
1018 network is referred to as CLGECL. Due to the constraint of identical local parameters, CLGECL
1019 is expected to be robust to gradient drift. For this competition, we propose DP-CLGECL, which
1020 introduced AdamW as a local update, client sampling, and Gaussian mechanism in DP for CLGECL,
1021 as summarized in Alg. 2.

1022 To follow the regulation of this competition task, we specified this operation as follows: First, we
1023 assume that each client's data set $D_k$ is partitioned into a set $\mathbb{G}_k$ of disjoint and pre-defined groups,
1024 and each client has different groups. The server randomly selects a subset $\mathbb{K}$ of $n$ clients in each
1025 round to update the global model. Each client receives the global model from the server for each
1026 round. The client selects a random subset $\mathbb{M}$ of groups, calculates the gradient $\Delta w_t^G$ by SGD with
1027 momentum for each group, and the gradient $\Delta w_t^G$ is updated with the dual variables $\lambda$, clipping it
1028 into clipped the gradient $\Delta \hat{w}_t^G$ to have a bounded $L_2$ norm of $S$, where $S$ denotes the sensitivity
1029 of the gradient $\Delta w_t^G$. The sum of $\Delta \hat{w}_t^G$ for all groups is calculated and perturbed by the Gaussian
1030 mechanism. Finally, the $k$ clients selected by the central server calculate the model update difference
1031 $w' - w_{t-1}$, send it to the server, and update the dual variable $\lambda$.

---

**Algorithm 2** Update rules of DP-CLGECL

---

1: Server:
2: Initialize common model $w_0$
3: **for** $t = 1$ to $R$ **do**
4:     Select set $\mathbb{K}$ of clients randomly
5:     **for** each client $k$ in $\mathbb{K}$ **do**
6:         $u_t^k = \text{Client}_k(w_{t-1})$
7:     **end for**
8:     $w_t = w_{t-1} + \frac{1}{|\mathbb{K}|} \sum_k u_t^k$
9: **end for**
10: Output: Global model $w_t$

---

11: $\text{Client}_k(w_{t-1})$:
12: $\mathbb{G}_k$ is a set of predefined disjoint groups of records in $D_k$
13: Select $\mathbb{M} \subseteq \mathbb{G}_k$ randomly
14: **if** t == 1 **then**
15:     Randomly initialize $\lambda_0$
16: **else**
17:     $\lambda_{t-1} \leftarrow \lambda_{t-2} + w_{t-1} - w'_{t-2}$.
18: **end if**
19: **for** each group $G$ in $\mathbb{M}$ **do**
20:     $w' = w_{t-1}$
21:     $\Delta w_t^G = \text{AdamW}(G, w', T_{gd}) - w_{t-1} + \lambda_{t-1}$
22:     $\Delta \hat{w}_t^G = w_t^G / \max(1, \frac{\|w_t^G\|_2}{S})$
23: **end for**
24: $w'_t = w_{t-1} + \frac{1}{|\mathbb{M}|} \left( \sum_G \Delta \hat{w}_t^G + \mathcal{N}(0, \mathbf{I}\sigma^2) \right)$
25: Output: Client model $w'_t - w_{t-1}$

---

**Privacy analysis**: In the privacy analysis of DP-CLGECL, we aim to determine $\varepsilon$ and $\sigma$ that ensure
1033 that $\Delta w_t^G + \mathcal{N}(0, \sigma^2 \mathbf{I})$ guarantees $(\alpha, \varepsilon)$-RDP. We then apply the composition on the RDP, and

convert the RDP to DP. The privacy analysis of FL-GROUP-DP[Marathe and Kanani, 2022, Galli et al., 2023] demonstrates a a method to guarantee $(\alpha, \varepsilon)$-RDP for $\Delta w_t^G + \mathcal{N}(0, \sigma^2 \mathbf{I})$. This analysis can be applied to our FL-GROUP-DP.

DP-CLGECL can guarantee $(\varepsilon, \delta)$-DP if $\sigma$ is used, satisfying the following

$$\varepsilon = \min_{\alpha} \left( R \cdot \xi_{\mathcal{N}}(\alpha \mid q) + \log((\alpha - 1)/\alpha) - (\log \delta + \log \alpha)/(\alpha - 1) \right), \tag{A8}$$

where

$$\xi_{\mathcal{N}}(\alpha \mid q) = \begin{cases} \dfrac{1}{\alpha - 1} \log \left( \displaystyle\sum_{k=0}^{\alpha} \binom{\alpha}{k} (1-q)^{\alpha-k} q^k \exp\left( \dfrac{k^2 - k}{2\sigma^2} \right) \right), \quad (\text{Integer}\,\alpha), \\[2em] \dfrac{1}{\alpha - 1} \log \left( \displaystyle\sum_{k=0}^{\infty} \dfrac{\Gamma(\alpha + 1)}{\Gamma(k+1)\Gamma(\alpha - k + 1)} (1-q)^{\alpha-k} q^k \dfrac{1}{2} \exp\left( \dfrac{k^2 - k}{2\sigma^2} \right) \mathrm{erfc}\left( \dfrac{k - z_1}{\sqrt{2}\sigma} \right) \right) \\[2em] \quad + \dfrac{1}{\alpha - 1} \log \left( \displaystyle\sum_{k=0}^{\infty} \dfrac{\Gamma(\alpha + 1)}{\Gamma(k+1)\Gamma(\alpha - k + 1)} (1-q)^{k} q^{\alpha-k} \dfrac{1}{2} \exp\left( \dfrac{k^2 - k}{2\sigma^2} \right) \mathrm{erfc}\left( \dfrac{z_1 - k}{\sqrt{2}\sigma} \right) \right), \\[1em] \quad (\text{Fractional}\,\alpha). \end{cases}$$

and a group is sampled with a probability of $q = \frac{C \cdot |\mathbf{M}|}{\min_k |\mathbb{G}_k|}$, $C$ is probability of client sampling.

**Compared methods**: In our testing, we mainly compared: 1) the baseline method based on FedAVG and 2) DP-CLGECL. We also tested their variant versions, such as replacing AdamW with momentum.

**Experiment results**: The best ANLS for all $\varepsilon$ was achieved by DP-CLGECL. By tuning the hyperparameter, the baseline method given by the competition organizers was also able to achieve a higher ANLS than the baseline presented.

The ANLS of DPCLGECL was further improved by using momentum instead of AdamW, as shown in Fig. A.5. This could be due to the clipping radius not being well-matched with the stochastic gradient using AdamW. A larger clipping radius can degrade the performance due to noise, thus, it seems better to use momentum than AdamW. In this competition, mitigating the gradient drift with CLGECL was also effective in improving performance. However, calculating the stochastic gradient that matches the clipping radius was the most effective in improving performance.

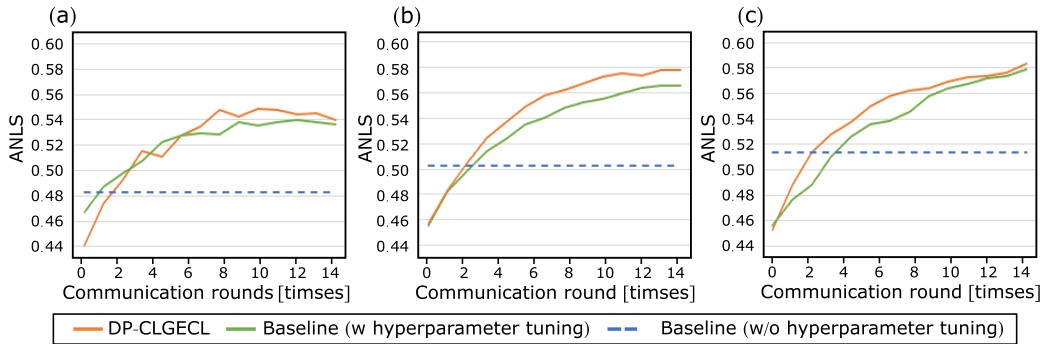

Figure A.4: convergence curve evaluating using the global model. (a) validation ANLS at $\varepsilon = 1$, (b) validation ANLS at $\varepsilon = 4$, (c) validation ANLS at $\varepsilon = 8$. We used clipping radius $S = 0.5$, the number of client sampling $C = 2$, the learning rate $\eta = 0.0002$, and the number of communication round $R = 14$ for hyperparameter selection.

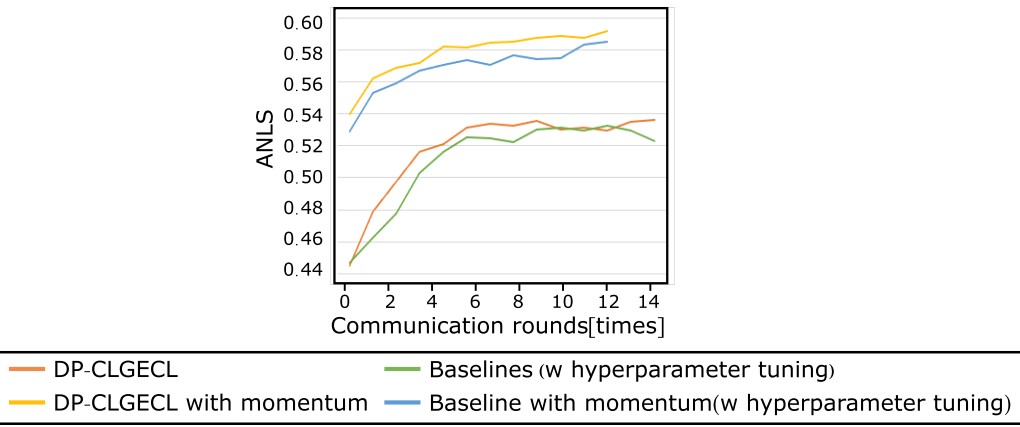

Figure A.5: convergence curve evaluating using the global model at $\varepsilon = 1$. (Left) Validation accuracy, (right) Validation ANLS. We used clipping radius $S = 0.5$, the number of client sampling $C = 2$, learning rate $\eta = 0.0004$, and the number of communication round $R = 12$.

