# OpenReview forum: "NeurIPS 2023 Competition: Privacy Preserving Federated Learning Document VQA"
_NeurIPS.cc/2024/Datasets_and_Benchmarks_Track — Submitted to NeurIPS 2024 Track Datasets and Benchmarks_

### Official Review · Reviewer_8VKs · 2024-07-09
**Detailed Summarization but Lacking Insights**

**Rating:** 4
**Confidence:** 3
**Correctness:** It is neither a dataset or a benchmar…
**Clarity:** Yes.

**Review:**

Strengths

S1. The topic of the competition is both timely and crucial, emphasizing the significant issues of communication and privacy in Federated Learning.

S2. The manuscript successfully details the competition's structure and the winners' solutions, providing a clear overview of the event.

Weakness

W1. While this manuscript serves effectively as a competition brief, it falls short of providing the strategic insights necessary to steer future research in Federated Learning (FL). A research paper of this nature should furnish profound insights that forecast and shape future directions in the field. Although some summaries are presented in Section 6, they require a more comprehensive analysis. Moreover, the focus of the analysis should extend beyond the competition's winners to include insights from other submissions as well, thereby enriching the overall findings.

W2. In regards to communication cost discussed in Track 1, the calculations are based on a synthetic model which may neglect various hidden communications and overlook significant overheads, such as socket headers. For a competition aimed at mimicking real-world federated learning scenarios, it is crucial to employ a genuine distributed implementation and provide a thorough evaluation that accurately reflects the complete communication overhead associated with federated learning.

**Strengths:**

See review.

**Additional Feedback:**

N/A

**Documentation:**

Yes.

**Limitations:**

Yes.

**Opportunities For Improvement:**

O1. The manuscript requires a more detailed investigation into all aspects of the competition entries, coupled with in-depth insights that can contribute meaningfully to the field of federated learning. Without such enhancements, the manuscript will remain a superficial recount, largely detached from the practical intricacies and challenges of effectively implementing federated learning systems.

**Relation To Prior Work:**

No. It is a summary of a competition.

**Summary And Contributions:**

The manuscript examines the outcomes of a federated learning competition, focusing on communication and privacy aspects. Although it provides detailed descriptions of the competition process and the winners' approaches, it lacks substantive insights that could significantly influence future research in federated learning. The analysis is superficial, primarily summarizing results without delving into the broader implications or offering a critical examination of all entries. This approach is insufficient for advancing the understanding necessary for real-world applications, where the complexities of communication costs and system overhead are crucial.

---

> ### Author Rebuttal · Authors · 2024-08-16
>
> >  In regards to communication cost discussed in Track 1, the calculations are based on a synthetic model which may neglect various hidden communications and overlook significant overheads, such as socket headers.
>
> Indeed, as you pointed out, we have not accounted for all potential costs in communication. However, it is important to note that our primary focus in the evaluation was not on assessing the system overhead but we focussed on the number of bytes required to transfer to model weights, which is why we chose to exclude it from the evaluation metrics. One possible solution to measure these additional costs could be using the gRPC communication mode implemented in the Flower framework. However, it is important to note that including socket headers would likely not significantly alter the overall evaluation compared to our current focus on measuring the total amount of transmitted model data. In other words, it would not lead to significant changes in the current evaluation.
>
> We will add a sentence to the manuscript mentioning this as a suggested improvement for future FL competitions and benchmarks.
>
> > The manuscript requires a more detailed investigation into all aspects of the competition entries, coupled with in-depth insights that can contribute meaningfully to the field of federated learning. Without such enhancements, the manuscript will remain a superficial recount, largely detached from the practical intricacies and challenges of effectively implementing federated learning systems.
>
>
>
> We will add the following discussions in the revised version.
>
> In Track 1, the winning approach achieves information transfer compression by leveraging LoRA and quantization techniques. In contrast, the runner-up approach modifies the local update rule to an adaptive optimization method, aiming for faster global model convergence. For the winning approach, the information compression drastically reduces the total amount of communication, with a 99 % reduction in data proving mode impactful than the up to 90 % reduction in the number of FL communication rounds. Since the approaches of these two teams can be integrated, incorporating the adaptive optimizer technique into a parameter-efficient fine-tuning (LoRA) appears to be a promising strategy to realize high-accuracy federated learning with moderate communication costs.
>
> In Track 2, the winning approach employs LoRA to reduce the number of parameters to be optimized. This may effectively mitigate the adverse effects of random noise on federated optimization. In contrast, the runner-up approach tackles noise disturbances through a variance reduction technique. Despite their distinct approaches, they achieved nearly identical overall accuracy across all privacy configurations. This suggests that combining these techniques could further enhance accuracy.

---

> > ### Comment · Reviewer_8VKs · 2024-08-27
> > **Response to Rebuttal**
> >
> > I thank author for their response. After reading and consideration, I decided to keep my original rating.

---

> > > ### Author Response · Authors · 2024-08-27
> > >
> > > Thanks for reading our rebuttal and your response.

---

### Official Review · Reviewer_HdXU · 2024-07-11
**Review for 1137**

**Rating:** 5
**Confidence:** 3
**Correctness:** The claims made in the submission cor…
**Clarity:** The paper is relevantly well-written.

**Review:**

Pros:
1. The competition addresses a real-world and challenging problem, which is insightful for future research.

2. The research problem is critical and interesting.

3. The paper is well-written.

Cons:
1. More quantitative performance and comparison from different perspectives (efficiency and robustness) are expected.

2. More basic solutions (e.g., existing foundation models) are expected for comparison. Only introducing two well-designed strategies is not insightful enough as a benchmark paper.

3. The technical difference between the challenge paper "[Tito et al., 2023b]" and this paper should be discussed. Is there any new benchmarking setting/scenario proposed in this paper?

**Strengths:**

Please see "Pros" in "Review".

**Additional Feedback:**

NA

**Documentation:**

Sufficient materials are given in the submission.

**Limitations:**

From my perspective, this paper doesn't have potential negative societal impact.

**Opportunities For Improvement:**

Please see "Cons" in "Review".

**Relation To Prior Work:**

Yes.

**Summary And Contributions:**

This paper introduce the competition of Privacy Preserving Federated Learning Document VQA (PFL-DocVQA). The datasets, evaluation, and task setting are introduced. The state-of-the-art solutions in the competition are given.

---

> ### Author Rebuttal · Authors · 2024-08-16
>
> > More quantitative performance and comparison from different perspectives (efficiency and robustness) are expected
>
> As you noted, we also recognize that efficiency and robustness are important performance indicators. However, as described in the accepted competition proposal the main metrics are communication cost (track 1) and privacy/utility trade-off (track 2).  Our competition focuses on these metrics and thus complements the other FL benchmarks that have a stronger focus on efficiency and robustness.
>
> > More basic solutions (e.g., existing foundation models) are expected for comparison. Only introducing two well-designed strategies is not insightful enough as a benchmark paper.
>
> We would like to point out to the reviewer as stated in the general response that this manuscript comprises the report of a competition that ran during NeurIPS 2023, following the competition setup that was proposed, reviewed and accepted in 2023. The aim of this manuscript is to analyze the results obtained from the competition.
>
> > The technical difference between the challenge paper "[Tito et al., 2023b]" and this paper should be discussed. Is there any new benchmarking setting/scenario proposed in this paper?
>
> We agree with the reviewer that the competition and the [Tito et al., 2023b] paper are related but we would like to point out the differences:
>
> - [Tito et al., 2023b] paper does not include any comments on the competition or the proposed solutions by the competition participants.
> - [Tito et al., 2023b] proposes a new membership inference attack particularly targeted at the provider level privacy to substantiate the problem of privacy leakage
> - [Tito et al., 2023b] has a slightly different baseline that outperforms all track 2 solutions in this manuscript due to more careful hyperparameter tuning. We are more than happy to include this baseline if the reviewer thinks that this would be a good idea.
> - Finally, the [Tito et al., 2023b] paper has been released after the competition has been concluded and can be seen as a concurrent but different manuscript.

---

> ### Comment · Area_Chair_usJx · 2024-08-30
>
> Thanks very much for your review. As the discussion is coming to a close, please check the authors' responses and provide your final comments, particularly any specific concerns you may have. Thank you again!

---

### Official Review · Reviewer_3pdw · 2024-07-20
**Privacy Preserving Federated Learning Document VQA**

**Rating:** 5
**Confidence:** 3
**Correctness:** Yes.
**Clarity:** Yes.

**Review:**

Strength:
1. This paper is clearly rewritten, the description of different components of this benchmark is clear and with great details.

Weakness:
1. There are many related FL benchmarks that evaluate communication costs and privacy. What’s the core difference between this benchmark vs these. It seems many benchmarks are focusing on more tasks to provide comprehensive evaluation. Also please provide enough background of related FL benchmarks. Similarly for related dataset.
2. The evaluation seems weak, only one model and one dataset is deployed.
3. For FL communication cost, what’s the setup for the network environment? Do we consider heterogeneity of network conditions for each FL client?
4. Section 6 seems weak, do you have any fundamental observations of this benchmark? What are the new perspectives the benchmark offers?

**Strengths:**

Please refer to the review section.

**Additional Feedback:**

Please refer to the review section.

**Documentation:**

Yes.

**Ethics:**

No.

**Limitations:**

Please refer to the review section.

**Opportunities For Improvement:**

Please refer to the review section.

**Relation To Prior Work:**

Please refer to the review section.

**Summary And Contributions:**

This paper introduces a benchmark on Privacy Preserving Federated Learning Document VQA to evaluate the privacy, model performance and FL costs

---

> ### Author Rebuttal · Authors · 2024-08-16
>
> > On related FL benchmarks that evaluate communication costs and privacy
>
> As you stated, many general FL benchmarks exist, but the situation for FL benchmarks with formal differential privacy in particular on a complex task like DocVQA is very different. We are not aware of a similar effort - and this is also the reason why our benchmark was accepted as a NeurIPS’23 competition.
>
> In more detail, there are plenty general FL benchmarks, such as
> - [Wei et al., 2023] FedAds: A Benchmark for Privacy-Preserving CVR Estimation with Vertical Federated Learning. In SIGIR 2023 Track on Resource and Reproducibility, 2023.
> - [Han et al., 2023] FedSecurity: A Benchmark for Attacks and Defenses in Federated Learning and Federated LLMs, arXiv:2306.04959, 2023.
> - [Zhang et al., 2023] FEDLEGAL: The First Real-World Federated Learning Benchmark for Legal NLP. In ACL (volume 1: long papers), 2023.
> - [Terrail et al., 2022] FLamby: Datasets and Benchmarks for Cross-Silo Federated Learning in Realistic Healthcare Settings, NeurIPS 2022 Track on Datasets and Benchmarks.
> - [Ye et al., 2024] FedLLM-Bench: Realistic Benchmarks for Federated Learning of Large Language Models, arXiv:2406.04845, 2024.
> - [Zhang et al., 2024] FLHetBench: Benchmarking Device and State Heterogeneity in Federated Learning, IEEE/CVF CVPR 2024..
> - [Lai et al., 2022] FedScale: Benchmarking Model and System Performance of Federated Learning at Scale, ICML 2022..
>
> Regarding privacy preserving-FL benchmarks, notable examples include FedAds and FedSecurity. FedAds focuses on privacy protection in vertical federated learning, which typically involves relatively small-scale model training. FedSecurity, on the other hand, includes components that simulate attack and defense mechanisms during FL training. However, it does not include datasets designed for real-world use cases or datasets with multi-modal data.
>
> For FL benchmarks that focus on real-world use cases, there are several benchmarks, such as [Terrail etal., 2022], [Zhang et al., 2023], [Ye et al., 2024]. Since [Terrail et al., 2022] and [Zhang et al., 2023] follow general federated learnings, they do not incorporate any formal privacy protection.
> In contrast, [Ye et al., 2024] performs benchmark tests involving Differential Privacy, though it primarily combines DP with FedAvg with a focus on LLMs while we are focussing on multi-modality with DocVQA. Furthermore, this benchmark is more recent than our competition and concurrent work with our report.
>
> In terms of measurement of communication costs during FL training, [Lai et al., 2022] introduced FedScale. FedScale evaluates the communication costs in cross-device FL, typically involving relatively small model training suitable for mobile devices.
>
> > What’s the core difference between this benchmark vs these.
>
> Our PFL-DocVQA is the first DocVQA competition designed to evaluate federated learning of large models in real-world scenarios while addressing realistic privacy concerns. Contributing to this competition demands advanced expertise in Differential Privacy, FL with large models, and DocVQA tasks. Developing a sophisticated method for this competition is essential for paving the way for the successful development of FL in real-world applications.
>
> Additionally, the privacy concerns and data distribution scenarios are thoughtfully designed. Specifically, each data provider is treated as a unit of privacy, with the assumption that each client collects invoice data from a different provider. This approach effectively mirrors realistic data heterogeneity.
>
> >  It seems many benchmarks are focusing on more tasks to provide comprehensive evaluation. Only one model and one dataset is deployed.
>
> We would like to point out to the reviewer as stated in the general response that this manuscript comprises the report of a competition that ran during NeurIPS 2023, following the competition setup that was proposed, reviewed and accepted in 2023. The accepted proposal states the focus on one dataset (specifically designed for the tasks) and one SOTA DocVQA model.
>
> Note that performing federated learning on DocVQA demands substantial computation resources (baseline by the organizer took over a day to complete a single hyper-parameter configuration, even with a SOTA GPU like RTX A6000).
>
> >  FL communication cost, what’s the setup for the network environment? Do we consider heterogeneity of network conditions for each FL client?
>
> We do not consider network heterogeneity among clients. This task (PFL-DocVQA) centers on cross-silo FL, with the assumption that clients are industrial companies and/or government institutions. Given that these clients typically have stable network connections due to their backbone networks, we believe that significant network heterogeneity is unlikely to occur. This network configuration is also commonly employed in other benchmark tests, such as [Terrail et al., 2022], which addresses cross-silo FL benchmarking in the healthcare domain.

---

> > ### Author Rebuttal · Authors · 2024-08-16
> >
> > > Fundamental observations of this benchmark? What are the new perspectives the benchmark offers?
> >
> > We will add the following discussions in the revised version.
> >
> > In Track 1, the winning approach achieves information transfer compression by leveraging LoRA and quantization techniques. In contrast, the runner-up approach modifies the local update rule to an adaptive optimization method, aiming for faster global model convergence. For the winning approach, the information compression drastically reduces the total amount of communication, with a 99 % reduction in data proving mode impactful than the up to 90 % reduction in the number of FL communication rounds. Since the approaches of these two teams can be integrated, incorporating the adaptive optimizer technique into a parameter-efficient fine-tuning (LoRA) appears to be a promising strategy to realize high-accuracy federated learning with moderate communication costs.
> >
> > In Track 2, the winning approach employs LoRA to reduce the number of parameters to be optimized. This may effectively mitigate the adverse effects of random noise on federated optimization. In contrast, the runner-up approach tackles noise disturbances through a variance reduction technique. Despite their distinct approaches, they achieved nearly identical overall accuracy across all privacy configurations. This suggests that combining these techniques could further enhance accuracy.

---

> ### Comment · Area_Chair_usJx · 2024-08-30
>
> Thanks very much for your review. As the discussion is coming to a close, please check the authors' responses and provide your final comments, particularly any specific concerns you may have. Thank you again!

---

### Official Review · Reviewer_EfAD · 2024-07-24
**Review for Privacy preserving FL competition**

**Rating:** 7
**Confidence:** 3
**Correctness:** Yes
**Clarity:** Yes, the paper is highly well-organiz…

**Review:**

Pros:
* The paper comprehensively summarizes the tasks, expectations, and how the competition works.

* After explaining each track, the authors describe their corresponding winner and runner-up algorithms.

* Section 6 is dedicated to the lessons learned, which is helpful for getting a better intuition for real-world applications of the competition and paves the way for similar competitions in the future.

Cons:
* The authors could explain how they decided on the FL setting; it would be more helpful if they could explain the dataset statistics and maybe select a few FL settings where we could better understand the impact of heterogeneity, number of clients, or limited resources on the winning algorithm.

**Strengths:**

Please check out the review section

**Additional Feedback:**

No

**Documentation:**

Yes

**Limitations:**

Yes

**Opportunities For Improvement:**

Please check out the review section

**Relation To Prior Work:**

Yes comprehensively

**Summary And Contributions:**

The paper reports a recent challenge for privacy-preserving document VQA using two important tools: federated learning and differential privacy. The underlying task is 'Document Visual Question Answering', and the goal is to improve the performance of a pre-trained model using collaborative learning.

The authors have properly motivated the benefits of the competition and explained the dataset and model with their different aspects.
For their evaluation, they have considered three of the most critical and important metrics in any FL setting: utility, communication cost, and privacy.  The competition had two tracks; the first track targets high utility and low communication cost, while the second track focuses on high utility with a limited privacy budget.

---

> ### Author Rebuttal · Authors · 2024-08-16
>
> > The authors could explain how they decided on the FL setting; it would be more helpful if they could explain the dataset statistics and maybe select a few FL settings where we could better understand the impact of heterogeneity, number of clients, or limited resources on the winning algorithm.
>
> We adjusted the FL setting in this paper to closely mimic realistic scenarios while also ensuring that the competition remains viable, given the participants’ computation resource limitations.
> In this PFL-DocVQA benchmark test, we chose to deploy a cross-silo FL benchmark targeting real-world use cases. We believe that automatic invoice processing is one of the most promising applications.
> The specific values of parameters for our FL setting are provided in lines 122–131 and Appendix A.
>
> Below, we outline the design principle for the following components.
> - (a) Heterogeneity: We assume that each client collects invoices from distinct providers (a.k.a. suppliers). This scenario results in a highly non-i.i.d. data distribution, reflecting real-world use cases and client-specific differences.
> - (b) The number of clients: We artificially set the number of clients, while taking the participants’ computation resource limitation. Note however that it is common to use the small number of clients in general cross-silo federated learning studies.
> - (c) Client resources: In the cross-silo federated learning, each client typically possesses significant computational power and a stable network connection, allowing for full participation in the entire training process.

---

> > ### Comment · Reviewer_EfAD · 2024-08-30
> > **Response to rebuttal**
> >
> > I wanted to thank the authors for their response, and I want to keep my rating.

---

> > > ### Author Response · Authors · 2024-08-31
> > >
> > > Thanks for reading our rebuttal and your response.

---

### Author Rebuttal · Authors · 2024-08-16

We would like to thank the reviewers for taking the time to review our competition analysis report. We replied to the individual remarks below but would like to clarify two general points regarding the call for a competition analysis and shaping the community:

**Competition analysis**

We would like to clarify that this manuscript comprises the report of a competition that ran during NeurIPS 2023, following the competition setup that was proposed, reviewed and accepted in 2023. The aim of this manuscript is to analyze the results obtained from the competition. Details of the benchmark and the setup of the competition are also available at the accepted competition proposal (https://github.com/rubenpt91/PFL-DocVQA-Competition/blob/master/framework_documentation/NeurIPS_2023_ELSA_Competition_Proposal.pdf). The competition followed the accepted proposal.

From https://neurips.cc/Conferences/2023/CallForCompetitions:

> This year, there are new publication requirements for competition reports co-authored by both organizers and participants. Unlike previous years, accepted competitions in 2023 will be required to submit their post-competition analyses as papers to the 2024 NeurIPS D&B track (next year). **In order to ensure that the competition proposal and results are consistent, reviewers in 2024 will have access to the 2023 proposals. To minimize experimental bias, any deviations from the proposal will need to be justified.**

**About the influence on the community and other FL benchmarks/competitions**

As the reviews noted, many general FL benchmarks exist, but the situation for FL benchmarks with formal differential privacy in particular on a complex task as DocVQA is very different. We are not aware of a similar effort - and this is also the reason why our benchmark was accepted as a NeurIPS’23 competition. We believe that by introducing differential privacy to the complex task of DocVQA we are shaping the future of the intersection of these communities, some follow-up work already noted the impact of the competition to the DocVQA community by making the community aware of the privacy challenges in DocVQA applications [Biescas et al., 2024] and DocVQA was adopted by privacy researchers as a new complex and realistic task [Wu et al., 2024] as discussed in Section 6.2.

[Biescas et al., 2024] Geocontrastnet: Contrastive key-value edge learning for language-agnostic document understanding. arXiv:2405.03104, 2024

[Wu et al., 2024] Privacy-preserving in-context learning for large language models, ICLR 2024

---

### Author Response · Authors · 2024-08-26
**Reminder of the rebuttal**

Dear reviewers,

as you are probably aware the discussion period for this track ends this week and we have not received any response to our rebuttal yet. We eagerly await your consideration regarding our rebuttal.

Furthermore, we are still eager to clarify any open questions or answer any further requests regarding our competition analysis if needed.

Best,
authors of paper 1137

---

### Decision · Program_Chairs · 2024-09-26

**Decision:**

Reject

**Comment:**

The paper reports a recent challenge for privacy-preserving document VQA using two important tools: federated learning and differential privacy. While some reviewers appear to have misunderstood the competition analysis criteria, the paper still presents several significant issues as highlighted by the reviewers:

1. It does not provide in-depth insights that could foster future research in federated learning.
2. The calculation of communication costs fails to account for significant overheads.